# Transforming Pharmacy Students’ Perceptions of Diabetes: An Innovative Teaching Approach Using Patient Interviews and Photovoice

**DOI:** 10.3390/pharmacy13030069

**Published:** 2025-05-20

**Authors:** Jenifer Santos, Manuel Machuca

**Affiliations:** 1Comprehensive and Multidisciplinary Sciences Applied to Health Group, Departamento de Ciencias de la Salud y Biomédicas, Facultad de Ciencias de la Salud, Universidad Loyola Andalucía, Avda. de las Universidades s/n, Dos Hermanas, 41704 Sevilla, Spain; jsgarcia@uloyola.es; 2Departmento de Ciencias Biomédicas y de la Salud, Faculty of Health Sciences, Universidad Loyola Andalucía, Dos Hermanas, 41704 Sevilla, Spain

**Keywords:** teaching, medication experience, experiential learning, diabetes, health anthropology, competency, pharmacy education

## Abstract

This study evaluated an innovative teaching methodology in a Health Anthropology course for Pharmacy students aimed at transforming their perceptions of diabetes. Through patient interviews and the photovoice technique, students gained deeper insights into the psychosocial aspects of the disease. Surveys were administered at the beginning, midpoint, and end of the course to assess shifts in students’ perspectives. The results revealed a significant change, with students evolving from a primarily biomedical view to a more holistic understanding, emphasizing the patient’s lived experience. Additionally, students responded positively to the innovative approach, citing enhanced motivation and learning, though some noted the methodology’s demanding workload. This teaching intervention fostered empathy and a broader perspective on the challenges faced by patients living with diabetes, supporting a more patient-centered and humanistic approach to healthcare. Despite the positive feedback, there was some uncertainty about whether this methodology could be applied to other subjects within the Pharmacy curriculum.

## 1. Introduction

Innovation in teaching is crucial for fostering an engaging, dynamic, and effective learning environment. As the world rapidly evolves, so too must the methods we use to educate future generations. Traditional teaching techniques, while foundational, often fall short of meeting the diverse needs of today’s students. Innovative teaching practices incorporate new technologies, interactive learning models and creative approaches to content delivery, all of which can better engage students, cater to different learning styles, and promote critical thinking [1]. By embracing innovation in education, we not only enhance the learning experience but also prepare students to adapt and thrive in an ever-changing world [2].

Nowadays, the interest and attention to the use of interactive methods, innovative technologies, and pedagogical and information technologies in the educational process is increasing day by day. In the process of education, the student becomes the main figure. Therefore, modern teaching methods—interactive methods and innovative technologies have a great role in training qualified professionals in higher educational institutions and faculties. As pedagogical innovation involving information and communications technology (ICT) may offer teachers the opportunity to create engaging learning environments in engineering courses [3], for other health science subjects, it is a challenge to improve the quality of teaching. In this study, patients’ interviews and photovoice were evaluated as innovative methodologies to incorporate into classroom lessons.

Photovoice is an innovative participatory method that integrates photography and storytelling to empower individuals and communities to capture and share their lived experiences. In the context of higher education, photovoice can be a powerful tool for increasing student engagement, critical thinking and reflection [4]. Using photography as a medium of expression, students are encouraged to visually represent their perspectives on academic topics, social issues or personal experiences. This approach fosters deeper connections between theory and practice, encourages collaborative learning and enables students to articulate complex ideas in a creative and accessible way [5].

In higher education, social science, education, public health and arts courses can be particularly effective in using photovoice [6,7]. It offers students the opportunity to develop their visual literacy, to communicate their points of view through non-traditional formats, and to engage with different perspectives within the classroom. It also enables students to address real-world issues and encourages them to think critically about their own role in society and the impact of their academic knowledge beyond the classroom. By incorporating photovoice into the university classroom, educators can create a dynamic and inclusive learning environment that encourages active participation and a deeper understanding of complex issues.

Patient interviews are an emerging and powerful teaching method in higher education, particularly within health and medical programs [8]. In this approach, students engage directly with patients to gain insight into their lived experiences, medical histories and perspectives on care. By using patient interviews as a teaching tool, educators aim to foster a deeper understanding of the human side of healthcare, going beyond textbooks to focus on the real-world application of medical knowledge. This methodology encourages active learning, empathy and critical thinking. Students are required to not only absorb clinical information but also interpret patient narratives in order to develop their communication skills and cultural competence [8]. These interactions provide students with invaluable firsthand experience in patient-centered care, enhancing their ability to approach medical practice holistically. Furthermore, patient interviews provide an opportunity for students to reflect on the complexity of healthcare systems and the diverse needs of individuals, making them an innovative tool for teaching professionalism, ethics, and interpersonal skills in healthcare education.

The subject of Health Anthropology has recently been introduced to the curricula of select Pharmacy degrees in Spain, including those offered by Loyola Andalucia University. The objective is to facilitate comprehension of the personal and social implications of illness for patients, which largely inform their attitudes and behaviors toward the illness and its treatment [9].

Health science students often share the same biases toward patients as healthcare professionals, who frequently underestimate patients’ experiences, fears, and expectations. Therefore, it is important for the instructors of this subject that students will be able to understand and respect the patient’s perspective, which is the best starting point for any healthcare practice. To achieve this, it is essential not only to teach basic concepts but also to ensure that learning is practical and begins with understanding the student’s own perspective.

Pharmacy students face the challenge of helping to minimize one of the most significant current public health issues: drug-related morbidity and mortality. They can do this by using an important healthcare tool known as Comprehensive Medication Management Services (CMM), which aims to identify, prevent and solve medication-related problems based on pharmacotherapy knowledge and an understanding of the patient’s medication experience—a concept that encompasses the patient’s knowledge, attitudes, expectations and fears regarding their illness and medications [10]. With this goal in mind, Loyola Andalucía University has designed the Medical Anthropology course to help students understand what it means for a patient to be ill. The course focuses on a highly prevalent disease with biomedical, psychological and social implications: diabetes.

Diabetes is a disease with a high prevalence worldwide, with serious consequences for the health of those who suffer from it and which causes enormous healthcare costs. Not only is it a major cardiovascular risk factor, but it also presents other complications related to macro and microcirculation in various tissues, which can lead not only to cardiovascular and cerebrovascular accidents but also to other consequences such as retinopathy and diabetic blindness, amputations, etc. [11,12]. Despite the availability of many oral and subcutaneous medications for its adequate control, diabetes is a disease whose approach has social and cultural implications that force sufferers to lead a personal and social life very different from that of the rest of society in terms of dietary limitations or the need for physical exercise. Hence, its real complications are much more important than expected due to the variety of pharmacological treatments [13,14]. This is why, from a teaching point of view, working with patients suffering from this disease can be a great opportunity for students of Health Sciences degrees, such as Pharmacy, to learn about the psychosocial dimension of diseases. This is, as proposed by the Anthropology of Health, an essential aspect for understanding the attitudes, behavior or expectations of patients in relation to their illness, which in anthropological terms is called the experience of illness and, with respect to medicines, the pharmacotherapeutic experience. These two concepts are fundamental for understanding the humanization of health as something that goes far beyond a paternalistic view of empathy toward patients and which consists of recognizing and knowing that, in addition to a biomedical perspective of the disease, understood as signs, symptoms and quantifiable parameters, any disease has another aspect, that of the patient, and which has to do with its psychological and social meaning, that is, how it influences their personal perception and how it interferes with their social relationships.

For this reason, this study evaluated a new teaching methodology to change students’ perception of a disease (diabetes) so that students understand the psychological and social dimensions of the disease focused on people who suffer from it. This was conducted using patient interviews, photovoice and questionnaires.

## 2. Methodology

### 2.1. Teaching Intervention Context

The subject “Health Anthropology”, which is part of the second-year curriculum in Pharmacy degree at the Health Sciences Faculty at the Loyola Andalucia University University, includes 30 h of classroom lessons, including lectures, patient interviews and clinical sessions. The subject is divided into three modules: Introduction to Anthropology, Health Anthropology and Pharmacist Anthropology.

Sixty-seven students participated in this innovative project as part of the study group. Since this subject is part of the second year of undergraduate studies, the students’ ages ranged from 18 to 25 years.

### 2.2. Procedure of Teaching Innovation

Students training on patients’ interviews: the teacher interviewed various patients and recorded them. Then, the video was projected in class to discuss the process of the interview with a special focus on patient medication experience.

Recruitment: Patient recruitment was carried out at the discretion of the students, applying inclusion criteria restricted to individuals with a confirmed diagnosis of any type of diabetes for at least one year and without a direct familial relationship to the student, as such a relationship was considered potentially disruptive to the therapeutic relationship.

Conducting Interviews: Students were assigned to conduct interviews with patients diagnosed with diabetes. The aim of the interview was to explore the patients’ experiences, challenges, and perspectives related to living with diabetes. Each student prepared a set of open-ended questions to guide the conversation, encouraging the patient to share their personal narrative. These interviews lasted about 30 min.

Transcription: After the interview, students transcribed the entire conversation, capturing both the patient’s words and the emotions conveyed. This step is crucial for ensuring accuracy and for later reflection on the patient’s experience.

Phrase Selection: Once the interview is transcribed, students analyze the conversation and select a specific phrase that resonates with them on a subjective level. This phrase should reflect something meaningful or significant that stood out during the interview, whether it related to the patient’s emotional experience, a challenge they faced or a moment of insight.

Photo Creation: Based on the selected phrase, students created a photograph that visually represented the essence of the phrase. This photo captured the student’s interpretation of the patient’s experience, translating the verbal narrative into a visual form that communicated the underlying emotions or themes. This component of the study was not included in the current manuscript, as we deem its significance merits separate and focused dissemination.

This process encouraged students to engage deeply with the patient’s story, fostering both empathy and creativity in their approach to understanding and representing the lived experience of diabetes.

### 2.3. Measurements of Perception About Diabetes by Students

In order to quantify the perception of diabetes by students, the following methodology was carried out:

On the first day of class, prior to any classes, students were invited to voluntarily complete an anonymous, non-graded diabetes knowledge survey. This survey included biomedical and psychosocial aspects of the disease, with responses given in five numerical categories, ranging from 1 (strongly disagree) to 5 (strongly agree). The survey was repeated mid-semester and at the end of the course to assess the student’s level of internalization of the concepts. These are the questions of the initial survey:

Question 1. The most important thing for a patient to feel good with diabetes is to have adequate blood glucose levels.

Question 2. If a patient feels distressed about diabetes, they should see a psychologist.

Question 3. The goal of communication with the patient is their health education.

Question 4. The main role of the healthcare professional, if diabetes is not controlled, is to make the patient aware of the consequences.

Question 5. A long-term diabetic patient with poor control is more difficult to help than a newly diagnosed patient.

Concerning question 2 (“If the patient feels distressed about diabetes, the patient should see a psychologist”), results for the pretest and the questionnaire at intermediate were obtained. However, this sentence seems to be confusing, and we decided not to use it any more. To replace it, we use question 6.

Question 6. If the patient feels distressed about diabetes, it is important to understand their medication experience to offer the best solution.

### 2.4. Satisfaction Survey

On the day of the final exam, students who wished to do so completed an anonymous satisfaction survey regarding the training they received, in addition to the survey conducted independently by the university to ensure the quality of the teaching provided.

### 2.5. Statistical Analysis

Satisfaction scores were analyzed as ordinal data obtained from a 4-point Likert-type scale (“strongly disagree” to “strongly agree”). Frequencies and percentages were reported to illustrate the distribution of responses across the scale.

### 2.6. Ethical Issues

This study was conducted in accordance with the Declaration of Helsinki. The interviewed patients received an information sheet about the work to be carried out and signed their consent. The confidentiality of patient data was rigorously maintained throughout the study, and participants were granted the unrestricted right to withdraw from the research at any stage without prejudice. Likewise, after completing the course and receiving their grades, the students agreed to allow their work to be used for research purposes, for which they signed an informed consent form. This research project was approved by the Loyola University Ethics Committee.

### 2.7. Study Limitations

The Pharmacy degree program at Loyola Andalucía University is still in its early stages. This study coincided with the inaugural offering of the Anthropology of Health course, and as such, no previous editions were available for benchmarking or comparative analysis.

## 3. Results and Discussion

Students were given a brief introduction to this novel methodology in the first class on the subject, and they answered the first survey. The impact of teacher intervention has been assessed in several ways. First, the changes in the answers to the questionnaire about the perception of diabetes between the beginning and the end of the academic course. The results of the questions asked in the initial, intermediate and final questionnaires are observed in Figure 1A–E for questions 1, 3, 4, 5 and 6, respectively. The bar chart illustrates the distribution of responses to the specific question at three different stages: the beginning, the intermediate, and the final phases. The responses are categorized into five levels of agreement: Strongly Disagree, Disagree, Neutral, Agree and Strongly Agree. Figure 1A shows the modification of the perception of the students about the following sentence: “Having adequate blood glucose levels is the most important thing for a person with diabetes to feel well”. Interestingly, at the end of the semester, students agreed with this sentence. This perception was modified over time thanks to the teacher’s intervention and his teaching methodology, with more than 60% of students agreeing to below 10%. This fact is very important since students have realized that diabetes is more than controlling glucose levels.

Figure 1B illustrates the variation of the perception of the students about the sentence: The purpose of communication with the patient is their health education. In the initial stage, more than 30% of the students think that the sentence is totally right. This fact changes over time and, at the end of the course, only about 15% of the students totally agree with this.

Figure 1C shows the perception of the students about the following sentence: “The primary role of the healthcare professional, if diabetes is not controlled, is to make the patient aware of the consequences” at the beginning, middle and end of the course. At the beginning of the course, more than 50% of the students agreed with statement four about the main role of healthcare professionals. However, at the end of the course, just 35% of them agree with that.

Figure 1D illustrates the perception of the students during the course about the following statement: A long-term diabetic patient with poor control is more difficult to help than a newly diagnosed patient. At the beginning phase, the highest percentage of responses is in the “Strongly Agree” category, followed by “Agree”. Fewer responses fall into the “Neutral” and “Disagree” categories, with minimal responses in “Strongly Disagree”. During the intermediate phase, there is a noticeable shift with an increased percentage of “Agree” responses, making it the most frequent. The “Disagree” category also sees an increase, while “Strongly Disagree” and “Neutral” have low representation. By the final phase, the distribution is more balanced across the “Agree” and “Strongly Agree” categories, with a significant decrease in “Disagree” responses. “Neutral” responses slightly increase, while “Strongly Disagree” remains the least frequent. Overall, Figure 1D suggests a general trend toward greater agreement with the statement in question 5 as time progresses, though some respondents maintain neutral or disagreeing views, especially during the intermediate phase.

Concerning question 6 (“If the patient feels distressed about diabetes, it is important to understand their medication experience to offer the best solution”), Figure 1E is observed. This figure shows the influence of time on the agreement of the students with this statement. The majority of responses during both the intermediate and end phases are concentrated in the “Strongly Agree” category, with both phases showing almost identical percentages. Nevertheless, the “Agree” category shows a higher percentage in the intermediate phase than in the final phase. Overall, the figure suggests strong agreement with the statement in question 6, particularly in the final phase, with little to no disagreement throughout both phases.

These figures prove that this innovative methodology (patient interview, photovoice) is able to modify the perception of a pathology such as diabetes in a short period of time. Interestingly, photovoice methodology has been used in nursing education with similar results [7].

A satisfaction questionnaire was completed by the students at the end of the Health Anthropology course. Table 1 shows the results as average scores for each question in the survey. In general, the use of this innovative teaching method (patient interviews and photovoice) was rated very positively by pharmacy students. They also emphasized that this teaching methodology increased their motivation and learning process. Finally, the students did not have a clear idea about whether this technology could be used in other courses.

Cloud maps are used and illustrated in Figure 2A,B to obtain a deeper insight into the opinions of the students. Figure 2A shows what students liked more about the methodology, and Figure 2B shows what they liked less. On the one hand, students liked more that this is a practical methodology that works with patients. On the other hand, they dislike the amount of work (interviews, photovoice) that they have to do. It is important to highlight that most of them do not see disadvantages in this methodology.

## 4. Conclusions

The study demonstrates that an innovative teaching methodology, which integrates patient interviews and the photovoice technique, significantly transforms the perceptions of pharmacy students regarding a disease, namely diabetes. As the course progressed, students demonstrated an increasing recognition of diabetes as a complex condition with multiple facets. By the conclusion of the course, the majority of students had moved beyond the narrow perspective of glucose control as the primary determinant of well-being for diabetic patients, instead recognizing the significance of emotional and social factors. This shift was evidenced by the survey results, which demonstrated that student opinions underwent a transformation in response to the interactive teaching methods employed. Furthermore, the use of photovoice, a methodology that integrates photography and narrative, enabled students to express the psychosocial challenges encountered by diabetic patients in an innovative and creative manner. This exercise facilitated not only the development of empathy but also encouraged critical reflection on the importance of these non-biomedical aspects in the context of healthcare.

In addition, this new teaching methodology was met with a positive response from the students, particularly in terms of enhanced motivation and a more profound engagement with the subject matter. This suggests that the approach not only succeeded in expanding the students’ comprehension of diabetes but also made the learning experience more dynamic and relevant to real-world applications. Although the methodology was perceived as highly efficacious within the context of the Health Anthropology course, some students expressed reservations about its potential integration into other subjects within the pharmacy curriculum. Finally, although the methodology was well received, a common concern among students was the workload involved, particularly in balancing the demands of conducting interviews, transcribing interviews and creating photovoice projects.

## Figures and Tables

**Figure 1 pharmacy-13-00069-f001:**
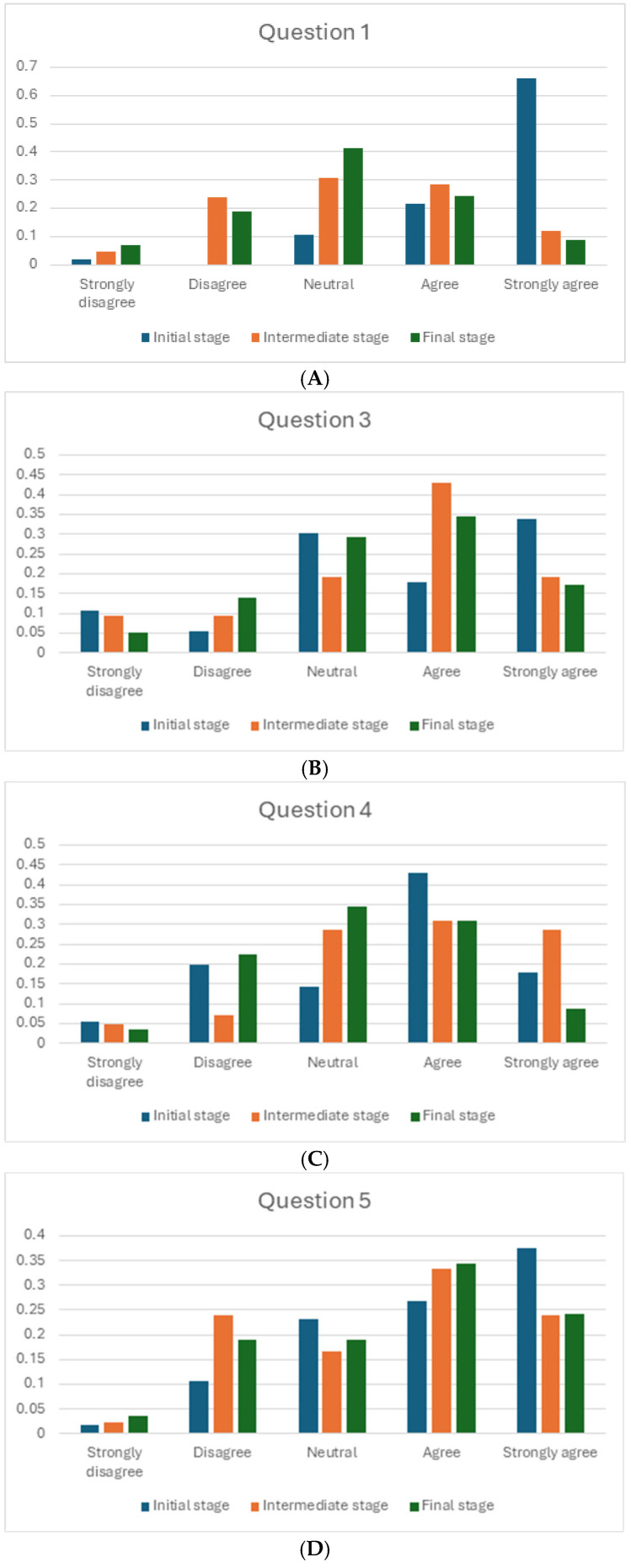
(**A**) Opinion of the students about question 1 at the initial, intermediate and final stages during the Anthropology course. (**B**) Opinion of the students about question 3 at the initial, intermediate and final stages during the Anthropology course. (**C**) Opinion of the students about question 4 at the initial, intermediate and final stages during the Anthropology course. (**D**) Opinion of the students about question 5 at the initial, intermediate and final stages during the Anthropology course. (**E**) Opinion of the students about question 6 at the initial, intermediate and final stages during the Anthropology course.

**Figure 2 pharmacy-13-00069-f002:**
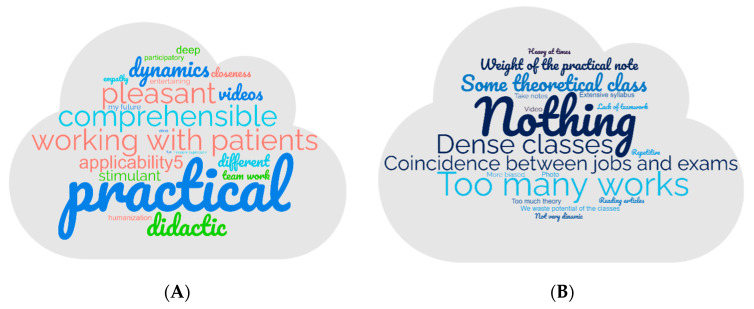
(**A**): Cloud maps about students liked most concerning the innovative methodology used. (**B**): Cloud maps about students liked least concerning the innovative methodology used.

**Table 1 pharmacy-13-00069-t001:** Average mark of each question of the satisfactory questionnaire carried out by the students.

Question	Average Mark
1 The use of this active teaching methodology has increased my motivation and participation in class.	4.3
2 The use of this teaching methodology has favored my learning process in Health Anthropology.	4.7
3. I prefer traditional teaching methods.	2
4. In general, I am happy with the use of this active teaching methodology for continuous assessment	4.5
5. I find it difficult to extrapolate this teaching methodology to other subjects in the degree.	2.7

## Data Availability

The original contributions presented in this study are included in the article. Further inquiries can be directed to the corresponding author.

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
