# Peer review of "Transforming Pharmacy Students’ Perceptions of Diabetes: An Innovative Teaching Approach Using Patient Interviews and Photovoice"

_pharmacy, 2025, doi:10.3390/pharmacy13030069_

Round 1
Reviewer 1 Report
Comments and Suggestions for Authors
This is an interesting paper that shares an innovative teaching method in a pharmacy school. The concepts of illness and medication experience must be well understood if pharmacists want to become patient-centered providers and contribute something new to the healthcare team. Thus, this manuscript is an illustration of what faculty can do to make learning more meaningful to students as they learn important concepts.
On line 93, I suggest to include the word "services" after Comprehensive medication management (CMM).
Also, on the methodology section, I suggest to put the full text in past tense, instead of future sense, like on line 145.
Author Response
All suggested modifications have been highlighted in red in the revised version of the manuscript.

Reviewer 2 Report
Comments and Suggestions for Authors
This is an attractive article and I hope it will be published soon. In general, healthcare disciplines need this kind of approach!!
However, I am concerned about a few things.
1. The procedure (2.2) is written in the future tense. Why? We would normally use past simple for something that you have really done.
2. I think a colon is missing in the first line of 2.2.
3. You mention phrase selection and photo creation in the procedure, but you do not talk about them in the results. If (as I imagine) you are going to use these in another paper, you could at least mention this and refer to your other paper. Otherwise we are left wondering what happened!
4. I find your reference to ethics slightly too off-hand. Please include more details (e.g. about anonymisation, etc.).
5. And the BIG thing that is missing here is a description of the sample of patients. Please include this! Obviously not details that would reveal identity, but we need to know how your students were able to contact this sample, what inclusion criteria were used, etc.
Author Response

(The authors gave the same response as above.)

Reviewer 3 Report
Comments and Suggestions for Authors
Thank you for submitting your manuscript to the journal 'Pharmacy'. The focus on multifactorial disease like diabetes is very relevant in pharmacy education. This study uses an innovative and interesting teaching methodology using patient interviews and photovoice to modify pharmacy student's perception of pathology of diabetes.
The methodology section is very brief. It will be helpful to elaborate this section and give details regarding the data analysis and statistical methods used. Is there any way to compare this data with a control group (e.g. with students from the previous cohort who were not exposed to this type of teaching methodology). Please elaborate on any limitations of this study.
Author Response

(The authors gave the same response as above.)
